# Role of SGLT2 Inhibitors, DPP-4 Inhibitors, and Metformin in Pancreatic Cancer Prevention

**DOI:** 10.3390/cancers16071325

**Published:** 2024-03-28

**Authors:** Tooba Laeeq, Maheen Ahmed, Hina Sattar, Muhammad Hamayl Zeeshan, Meher Binte Ali

**Affiliations:** 1Internal Medicine, University of Nevada, 4505 S Maryland Pkwy, Las Vegas, NV 89154, USA; 2Internal Medicine, Dow University of Health Sciences, Mission Rd., New Labour Colony, Karachi 74200, Pakistan; maheen_ahmed24@live.com (M.A.); hamayl32@gmail.com (M.H.Z.); 3Internal Medicine, University of Maryland Medical Center, 827 Linden Ave., Baltimore, MD 21201, USA

**Keywords:** diabetes, pancreatic cancer, anti-diabetic drugs, risk, prevention

## Abstract

**Simple Summary:**

Diabetes is an important risk factor for the development of pancreatic cancer due to the production of cytokines that cause increased cell proliferation. In this review, we have explored the effects of SGLT2 inhibitors, GLP-1 RA, DPP-4 inhibitors, and metformin on the risk of developing pancreatic cancer. Our review shows that some antidiabetic drugs may have a role in decreasing the risk of pancreatic cancer through different mechanisms. We have described their immunomodulatory roles, safety profiles, and association with pancreatic cancer.

**Abstract:**

Pancreatic carcinoma is a highly aggressive tumor that usually presents when it has already metastasized. Therapeutic options for cure remain scarce and rely on combination chemotherapy with limited sustainability. Diabetes is considered an important risk factor for the development of pancreatic cancer due to the production of proinflammatory cytokines, which result in increased cell proliferation. More than half of patients diagnosed with pancreatic cancer eventually develop diabetes due to the destruction of insulin-producing cells. The interlinkage of both diseases might identify a possible preventative strategy for reducing the incidence of pancreatic carcinoma. This study reviewed the recent literature on the association between pancreatic cancer risk and SGLT2 inhibitors, GLP-1 RA, DPP-4 inhibitors, and biguanides. There are mixed data regarding the relationship between GLP-1 RA and DPP-4 inhibitors and pancreatic cancer, with some trials suggesting that they might increase the risk. In contrast, studies have mostly revealed that SGLT2 inhibitors have an antiproliferative effect on various tumors, such as liver, pancreatic, prostate, bowel, lung, and breast carcinoma, which might be due to their mechanism of blockage of reabsorption of glucose by cells, lowering the amount of available glucose for the growth of tumor cells. Metformin, the first-line agent for diabetes, has also been shown to be associated with decreasing pancreatic cancer risk and improving prognosis in those who already have the disease. Dedicated trials are needed to further delineate the association of antidiabetic drugs with the risk of pancreatic cancer in the general population, as previous studies have mostly focused on diabetic patients.

## 1. Introduction

Pancreatic cancer remains among the most dangerous cancers worldwide. It has a high mortality rate, with a 5-year relative survival rate of 11.5% [1]. The Global Cancer Observatory (GLOBOCAN) 2020 observed that the incidence of pancreatic cancer was at an estimated 495,773 in 2020 [2]. There are different types of pancreatic cancer, notably pancreatic ductal adenocarcinoma (PDAC, comprising 85% of the total cases) and neuroendocrine tumors (comprising 5% of the total cases) [1]. By the time of diagnosis, 50% of pancreatic cancer is already metastatic, and more than 80% of patients present with unresectable tumors. Moreover, the lack of suitable biomarkers means we cannot identify patients who gain the most from a particular form of treatment. Despite some promising results from several early-phase clinical trials assessing novel approaches over the last 20 years, these have not been confirmed by subsequent Phase III trials [3,4].

Treatment for pancreatic cancer consists of different modalities, including surgery, chemotherapy, immunotherapy, and radiation. Palliative care, especially for pain, is also an integral part of management [5]. Common chemotherapy agents used are FOLFIRINOX (5-Flurouracil, folinic acid, irinotecan, and oxaliplatin) and Gemcitabine (Gem) [6]. Neoadjuvant chemotherapy and radiation are performed in locally invasive pancreatic cancer to improve margin-negative resection rates. They also increase the number of patients eligible for surgery by shrinking the tumor [7]. Resection of the pancreas, along with adjuvant chemotherapy, is the only long-term treatment option for patients with pancreatic cancer. In the next few years, cases of PDAC will gradually increase. By 2030, it is estimated that PDAC could be the second leading cause of cancer-related deaths [8]. Therapeutic options remain scarce and mostly rely on combination chemotherapy with limited sustainability.

Diabetes is associated with an increased risk of pancreatic cancer. There is an increased risk of malignant transformation with high blood insulin and glucose levels [9]. Proinflammatory conditions produced by diabetes induce Interleukin-6 (IL-6), Tumor Necrosis Factor-α (TNF-α), and other chronic inflammatory markers [10]. IL-6 induces the STAT3 (signal transducer and activator of transcription 3) signaling pathway, resulting in pancreatic cancer cell proliferation. It also increases the release of Type 2 T helper cell cytokines and the extracellular signal-regulated kinase 2 (ERK2) signaling pathway, indicating that IL-6 promotes a tumor microenvironment for cancer cell proliferation [11]. Other pro-inflammatory cytokines, such as Interleukin-1a and Interleukin-1b, promote cancer growth and invasion.

The preferred treatment of choice for type 2 diabetes is oral antidiabetics. These drugs express their effects by the following mechanisms: (i) increasing the production of insulin by the pancreas; (ii) increasing the sensitivity of target cells to insulin; and (iii) reducing the reuptake of glucose by the intestine and kidneys [12]. It is observed that antidiabetic drugs from different classes have shown promising results in the treatment of cancer when combined with anticancer drugs. Agents such as biguanides, sulfonylureas, and thiazolidinediones have reported anti-proliferative effects on several cancer cells [13,14,15].

DPP-4 inhibitors (DPP-4 inhibitors) are oral antidiabetics that inhibit the enzyme DPP-4. This enzyme is expressed on the surface of most types of cells and can deactivate glucose-dependent insulinotropic polypeptide (GIP) and GLP-1 (Glucagon-like peptide 1). DPP-4 inhibitors express their actions by inhibiting the degradation of GIP and GLP-1. They help significantly reduce HbA1c levels in diabetic patients and prevent weight gain, and there is a low risk of hypoglycemia when taking this drug [16]. GLP-1 has proven to stop cell proliferation and kill colon cancer cells, demonstrating its protective character in the case of colon cancer [17]. Sitagliptin and Vildagliptin are the first DPP-4 inhibitors available. Sitagliptin is a high-potency DPP-4 inhibitor. It is observed that when sitagliptin is given chronically at therapeutic levels in rats, the risk of developing colon cancer decreases [18]. 

GLP-1 agonists for type 2 diabetes work by imitating the natural hormone GLP-1. They promote balanced blood sugar levels in several ways: they stimulate glucose-dependent insulin release, suppress glucagon secretion from β cells, and slow gastric emptying. In addition, these drugs suppress appetite and can cause weight loss as a desirable side effect. Gier et al. reported that administering GLP-1 RAs to diabetic patients would not raise a new threat of thyroid tumors [19]. Another piece of research has shown that GLP-1 RAs can produce a long inhibitory effect on prostate cancer by downregulating the PI3K/AKT pathway and mTOR signaling, in addition to suppressing cells from other common forms of pancreatic and prostatic disorders through modulation of NF-kB activity [20]. Likewise, GLP-1 RAs were shown to suppress the growth of breast and cervical cancer, making them promising candidates for treating such cancers [21]. Zhao et al. showed that liraglutide exerted significant anti-proliferation activity and pro-apoptotic effects against gemcitabine-resistant human pancreatic cancer cells [22]. Some think that liraglutide and exenatide can cause apoptosis and induce autophagy, which slows down endometrial cancer. Furthermore, exenatide was found to increase the expression of GLP-1 and suppress PI3K/AKT activity as a mechanism underlying its cytostatic actions on ovarian cancer cells in diabetic patients with this kind of malignancy [23]. Consequently, taking GLP-1 RAs together with an anticancer drug would have the effect of indirectly suppressing cell migration and invasion as well as tumor growth due to enhanced chemosensitivity. 

Sodium-glucose co-transporter 2 (SGLT2) inhibitors are a recent class of antidiabetics that have also been shown to exert an anticancer effect. These antidiabetics reduce glucose reabsorption in the proximal tubule via active reverse transport, mediated by the SGLT2 transporter, resulting in glucosuria, which reduces plasma glucose [24]. In vitro studies have demonstrated that SGLT2 inhibitors possess antiproliferative effects on various tumors, such as those of the liver, pancreas, prostate, bowel, lung, and breast [25]. This may be due to the decrease in the amount of available glucose for growth and metabolism in tumor cells [25]. Moreover, SGLT2 is not the only glucose transporter protein expressed in cancer cells. There are also other types whose levels may increase due to metabolic reprogramming by various tumors. Table 1 summarizes the mechanism of action of drugs and their role in pancreatic cancer.

## 2. GLP-1 Receptor Agonists

### 2.1. Background

GLP-1 is a glucoregulatory hormone secreted in response to meal intake. One of the many functions of GLP-1 is the “incretin effect” [35]. Diabetes has an insidious onset, with an imbalance between insulin secretion and insulin resistance, leading to a functional insulin deficit [36]. Due to their incretin effect, GLP-1 RAs were eventually investigated for their application in managing diabetes [35]. Other effects of note include delayed gastric emptying, inhibition of glucagon release, and reduced food intake [37,38,39]. Weight loss is a commonly desired outcome [40,41].

### 2.2. Immunomodulation

There has been growing information on the role of GLP-1 in regulating innate immunity. Shiraishi et al. found that exenatide plays a role in promoting macrophage activation toward the M2 phenotype [42]. The M2 phenotype helps dampen inflammation and plays a role in tissue repair [43]. Another study involving a GLP-1 analog, Lixisenatide, observed decreased atheroma plaque size in mice by promoting the M2 phenotype in macrophages [44]. In this context, many intratumor macrophages exhibit the M2 phenotype, linked to an unfavorable prognosis in many cancers. M2 macrophages contribute to tumor cell metastasis and proliferation through their anti-inflammatory mechanisms [45].

### 2.3. Pancreatic Cancer

A temporal relationship was suggested between pancreatic cancer and the use of GLP-1 RAs. Perfetti et al. found that GLP-1 induces cell proliferation in the pancreas and augments Beta cell mass in rats, suggesting a potential benefit in treating diabetes mellitus [46]. However, a different animal study observed premalignant changes in the histological specimen of a rat pancreas treated with exenatide [47].

To address the potential effect of incretin therapy on the endocrine and exocrine pancreas, Butler et al. procured pancreas from brain-dead organ donors with diabetes mellitus. These were subdivided into subjects who had received incretin therapy and those who had not. They noted that both exocrine and endocrine pancreas were markedly enlarged in patients on incretin therapy and premalignant changes. Furthermore, they found alpha cell hyperplasia, neuroendocrine tumors, and microadenoma formation, giving more weight to concerns regarding the proliferative actions of GLP-1 therapy and a possible increased risk of neuroendocrine tumors [48].

Elashoff et al. evaluated the US Food and Drug Administration (FDA) database from 2004 to 2009 and found that the risk of pancreatic cancer was 2.9-fold greater in patients taking exenatide compared to other therapies [49]. However, we must remember the various limitations of the FDA database, including incomplete data and reporting bias [49]. Yang et al. examined the FDA Adverse Events database from 2004 to 2020 to assess the relationship between GLP-1 RA and all types of neoplasms. They reported significant signals between GLP-1 RA use and certain tumors, including thyroid and pancreatic cancers. However, a causal relationship between the use of drugs and the adverse events reported could not be established [50].

GLP-1 RAs have been associated with pancreatitis in the past, and pancreatitis is a well-known risk factor for pancreatic cancer [49,51]. One of the many proposed mechanisms is the formation of gallstones secondary to weight loss associated with GLP-1 RAs. Gallstones can lead to gallstone pancreatitis [52]. Javed et al. recently reported a case of acute pancreatitis in a patient taking liraglutide for 20 months. The patient was advised to stop taking the said medicine and had no repeated bouts of pancreatitis in the 10-month follow-up [53]. Various other GLP-1 RA-induced pancreatitis cases have been reported [53].

Several meta-analyses have been carried out, and no association has been identified between the usage of GLP-1 RA and pancreatic cancer [26,27,54,55]. Aziz et al., in a recent meta-analysis, reported no elevated risk of pancreatic cancer with GLP-1 Ras. [1.05 (0.57–1.95); *p* = 0.87]. The mean duration of the trials included ranged from 1.3 years to 5.4 years [26]. Cao et al. combined data from 37 trials, with the primary endpoint being the identification of any cancer, and found GLP-1 RAs were not associated with an increased risk of cancer (OR 1.03 [95% CI 0.95–1.12]; *p* = 0.41) [27]. On analyzing individual drugs, they found albiglutide showed a notably lower cancer risk. (OR 0.76 [95% CI 0.60–0.97]; *p* = 0.03) [27]. Pancreatic cancer and the use of GLP-1 RAs showed no significant association (OR 1.05) [0.68–1.60] (*p* = 0.83) [27]. Monami et al. reported similar results. (0.94 [0.52–1.70], *p* = 0.84) [54]. Pinto et al. pooled 12 trials and found no significant association between GLP-1 RA use and pancreatic cancer. (OR 1.06; 95% CI 0.67 to 1.67). The mean follow-up duration in the trials included was 1.74 years [55].

### 2.4. Thyroid Cancer

The safety of GLP-1 RAs in patients with a possibility of thyroid cancer has not been established. Studies have identified that the use of GLP-1 RAs in rodents can lead to C-cell proliferation and tumorigenesis, secondary to GLP-1 receptor activation [56,57]. Knudsen et al. found that similar results were not replicated in non-human primates, which led us to believe that either the association is weak or maybe a longer course of treatment with GLP-1 RAs in humans might result in the unintended side effect [57]. Gier et al. studied thyroid tissue samples taken from patients with medullary thyroid carcinoma, C-cell hyperplasia, papillary thyroid carcinoma, and normal thyroids. The investigation revealed that the GLP-1 receptor was frequently expressed in all of them [19]. A recent analysis of the FDA Adverse Events database revealed a similar potential correlation [50]. While most meta-analyses were unable to detect a significant association between the use of GLP-1 RAs and any thyroid malignancies, a recent meta-analysis revealed that treatment with GLP-1RAs might be associated with a moderate risk of thyroid cancer [28,58,59]. The general recommendation is to not use GLP-1RAs in patients with a family or personal history of thyroid carcinoma or Multiple Endocrine Neoplasia 2A or 2B syndromes [60].

### 2.5. Safety Profile

The extended-term safety of GLP-1 RAs has not been conclusively established. 

Studies with prolonged follow-up periods are recommended to investigate further and ensure their safety profile. Gastrointestinal side effects are frequently encountered, including nausea, diarrhea, and vomiting [61]. Severe vomiting and diarrhea can also lead to acute kidney injury secondary to dehydration [62]. Animal studies have also reported cases of thyroid cancer secondary to GLP-1 RA use [57]. While the long-term safety of GLP-1 RA is still a matter of debate, multiple trials have demonstrated their efficacy in providing benefits for patients with cardiovascular disease and reducing overall mortality [63].

## 3. DPP-4 Inhibitors

### 3.1. Background

DPP-4 inhibitors are oral antidiabetic agents that inhibit the enzyme DPP-4, which is responsible for the breakdown of GLP-1 and GIP [64]. GIP is secreted from the enteroendocrine K cells and regulates glucose-dependent insulin release. GIP induces pancreatic beta-cell proliferation, promotes pro-insulin gene transcription and translation, and suppresses hepatic glucose production [65]. GIP receptors are also widely found in adipocytes, increasing glucose transport and fatty acid synthesis [65]. The actions of GLP-1 have been covered above.

### 3.2. Immunomodulation

DPP-4, also known as CD-26, is a membrane protein. In addition to the degradation of incretin molecules, it targets a range of other substrates, namely chemokines, including RANTES, MDC, Eotaxin, IP-10, and mig [66]. Besides its catalytic activity, it works as a ligand to bind various extracellular molecules [67]. One of the extracellular molecules it binds to is Adenosine Deaminase (ADA) [68]. ADA is responsible for the hydrolysis of adenosine to inosine, leading to low concentrations of Adenosine in the extracellular matrix. We know that high concentrations of adenosine are known to inhibit the proliferation of T lymphocytes. However, ADA bound to CD26 counteracts this inhibitory effect. Therefore, the CD26/ADA complex is important for T-cell activation [68,69]. Pacheco et al. found a 3-to-34-fold increase in the production of T helper cells and other pro-inflammatory cytokines through the ADA/CD26 co-stimulatory signal [69]. CD-26 also modulates B-cell and natural killer cell responses [70,71].

Kagal et al. studied rats to examine the effects of DPP-4 inhibitors, specifically vilagliptin and saxagliptin, on acute and subacute inflammation. Inflammation was induced, with edema serving as an acute inflammation indicator, and subacute inflammation was characterized by granuloma formation. The findings revealed significant anti-inflammatory properties for both drugs in acute and subacute models [72].

### 3.3. Tumorigenesis

DPP-4 inhibitors have been found to have complex effects on tumorigenesis. Mezawa et al. found that stromal CD-26 expression decreased in cancerous cells, while the noncancerous regions showed adequate staining for CD-26. This diminished CD-26 expression was associated with poor outcomes [73]. In an animal study, Yang et al. found that the inhibition of DPP-4 might be related to increasing the potential of metastatic breast cancer [74].

On the other hand, Ng et al., in a retrospective analysis, found that patients with colorectal cancer on DPP-4 inhibitors showed a markedly improved 5-year disease-free period [75]. In a study conducted by Chou et al., it was identified that while a low cumulative defined daily dose of DPP-4 inhibitors was associated with a decreasing risk of colorectal cancer, a high cumulative defined daily dose showed an increasing risk. The authors concluded that there was a J-shaped association and emphasized the need for further studies [76].

The role of DPP-4 and its inhibition have also been studied in the context of Hepatocellular Carcinoma (HCC). While Yu et al. reported that HCC cells with decreased expression of DPP-4 had poorer survival outcomes, Nishina et al. found that the group with high expression of CD26 in mice showed more advanced tumor stages and lower overall survival [77,78]. Bishnoi and co-authors analyzed the Surveillance Epidemiology and Endpoint Research (SEER)-Medicare database and found that DPP-4 inhibitors improved overall colorectal and lung cancer survival [79]. Ali et al. found that DPP-4 was associated with higher progression-free survival in patients with advanced airway and colorectal cancer [80]. It seems the tumor environment plays a role in influencing the activity and expression of DPP-4 [81]. So far, in cancer research, one area where DPP-4′s role has been frequently addressed is pancreatic cancer.

### 3.4. Pancreatic Cancer

As discussed earlier, GLP-1 proliferates the pancreas, potentially leading to premalignant changes [47]. DPP-4 is responsible for the breakdown of GLP-1, rendering it inactive [64]. When we use DPP-4 inhibitors, we prolong the half-life of GLP-1 in the serum [64]. Elashoff et al. found an increased risk of pancreatic cancer among patients using sitagliptin in their review of the FDA database [49]. In an animal study, Matveyenko et al. found that sitagliptin increased the risk of pancreatitis and ductal metaplasia [82]. Butler et al. reported similar results for sitagliptin as they did for exenatide in brain-dead organ donors [48]. In an investigation led by Gokhale et al., there was no elevated risk of pancreatic cancer among patients taking DPP-4 inhibitors. The brief duration and small sample size limited the trial [31]. Engel et al. found sitagliptin safe overall, with no increased risk of malignancy [30]. In a meta-analysis including 38 trials, Pinto et al. found no association between pancreatic cancer and the use of DPP-4 inhibitors (odds ratio 0.65; 95% CI 0.35–1.21) [29]. The duration of the trials included ranged from 24 weeks (about 5 and a half months) to 260 weeks (about 5 years) [29]. So far, there is no established risk of pancreatic cancer with DPP-4 inhibitors [83].

### 3.5. Thyroid Cancer

The role of DPP-4 inhibitors in thyroid cancer has recently gained popularity. Investigators found that DPP-4 is well expressed in tumor cells and is associated with metastasis and poorer clinical outcomes. They also suggested the use of sitagliptin as a possible drug to be further investigated for the treatment of thyroid cancer [84]. Another study implicated DPP-4 in the cell signaling axis of thyroid cancer metastasis; however, they were unable to see any beneficial effects of DPP-4 inhibition through sitagliptin [85]. Lee et al. found that DPP-4 expression was related to advanced tumor stage in papillary thyroid carcinoma and that DPP-4 inhibitors significantly reduced colony formation, cell migration, and invasion [86]. Meta-analyses have found no increased risk of thyroid cancer in diabetic patients taking DPP-4 inhibitors [87,88].

### 3.6. Safety Profile

Dpp-4 inhibitors do not affect all-cause mortality [89,90]. Similarly, they appear to exert a neutral effect on the progression of diabetic kidney disease [91,92]. Additionally, DPP-4 inhibitors were discovered to manage glycemic levels effectively without the associated risk of hypoglycemia [92]. The primary adverse effect clinicians must be vigilant about is acute pancreatitis [93].

## 4. SGLT2 Inhibitors

### 4.1. Background

There are two types of sodium-glucose transporters: SGLT1 and SGLT2. SGLT2 is solely expressed in the proximal convoluted tubules in the kidney. SGLT2 inhibitors lower glucose levels by blocking glucose reabsorption in the proximal convoluted tubules, thus reducing plasma glucose levels [94]. They also have effects on other cellular pathways. Studies have shown that they have anti-inflammatory properties. These effects of SGLT2 inhibitors are attributed to the reduction in pro-inflammatory markers, primarily IL-6, C-Reactive Protein (CRP), TNF-α, and Monocyte Chemoattractant Protein-1 (MCP-1), as well as fibrosis and apoptosis [95]. A meta-analysis of trials conducted on rodents revealed a decrease in IL-6 [standardized mean difference (SMD: −1.56, 95% CI −2.06 to −1.05), CRP (SMD: −2.17, 95% CI −2.80 to −1.53), TNF-α (SMD: −1.75, 95% CI −2.14 to −1.37), and MCP-1 (SMD: −2.04, 95% CI −2.91 to −1.17)] [96]. It also stimulates anti-inflammatory macrophages and results in an increase in anti-inflammatory cytokines such as IL-10. SGLT2 inhibitors also exhibit antioxidant properties [97]. They decrease oxidative stress by reducing the generation of Reactive Oxygen Species (ROS) and mitochondrial Ca^2+^ overload, which is triggered by glucose overload in cells. They amplify antioxidant pathways via superoxide dismutase and glutathione peroxidases and suppress pro-oxidants such as NADPH oxidase 4. Canagliflozin suppresses glycolysis by inhibiting the PI3K/AKT/mTOR pathway, decreasing cell growth and proliferation [98]. It also increases AMP-activated protein kinase (AMPK) activation, inhibiting cellular protein synthesis and inducing apoptosis [98]. It also induces apoptosis via caspase-3 activation by activating the BAX/Bcl-2 ratio [99]. These mechanisms make them potential anticancer agents. 

Glucose is a major energy source for rapidly growing tumor cells to fulfill their energy requirements. Some cancer cells overexpress glucose transporters, GLUT and SGLT, on their membranes to maintain high glucose concentrations for their high metabolic activity [25,100]. SGLT2 inhibitors also inhibit cell proliferation by reducing glucose uptake by cells. This reduces the production of ATP, which decreases cancer cell growth and proliferation [25].

### 4.2. SGLT2 Inhibitors and Pancreatic Cancer

SGLT2 inhibitors are effective in certain cancers that express SGLT2 transporters on their cell surfaces, such as pancreatic cancer. They decrease pancreatic cancer cell growth by suppressing glycolysis via the PI3K/AKT/mTOR pathway and decreasing glucose uptake and lactate production by decreasing mRNA levels of genes for GLUT-1 and lactate dehydrogenase A [101]. A study conducted on patients with diabetes mellitus type 2 (DM type 2) using SGLT2 inhibitors or DPP-4 inhibitors revealed that the rate of acute pancreatitis was significantly decreased in patients taking SGLT2 inhibitors than in those taking DPP-4 inhibitors [76]. Another study performed in Japanese patients with DM type 2 revealed that administration of SGLT2 inhibitors for greater than 180 days (about 6 months) has a positive association with lowering the risk of developing pancreatic cancer (adjusted odds ratio: 0.58, 95% confidence interval: 0.31–0.99) [32]. Another trial was carried out to evaluate the effect of SGLT2 inhibitors on pancreatic cancer xenografts in mice [33]. The treatment was divided into four groups: placebo, canagliflozin, gemcitabine, and canagliflozin plus gemcitabine. Canagliflozin plus gemcitabine was more beneficial in reducing the rate of tumor growth and increasing necrosis compared to placebo and gemcitabine alone, respectively. The results concluded that canagliflozin reduces tumor growth and promotes tumor necrosis, whereas gemcitabine only affects tumor growth. SGLT2 inhibitors, in combination with chemotherapy and radiotherapy, have been shown to decrease tumor cell proliferation and growth in patients with pancreatic cancer [33]. 

### 4.3. Safety and Efficacy of SGLT2 Inhibitors Prevention

In patients with DM type 2, the use of SGLT2 inhibitors may be associated with reducing the risk of developing pancreatic cancer and improving cardiovascular and renal outcomes [102]. However, they also increase the risk of genitourinary infections due to glycosuria, acute kidney injury, amputations, bone fractures, diabetic ketoacidosis, and hypotension [103]. Thus, monitoring blood glucose levels, renal function, and volume status can minimize risks and improve outcomes. 

## 5. Metformin

### 5.1. Background

Metformin, a biguanide, is the first-line medication for treating DM type 2. It works by lowering glucose production in the liver by suppressing gluconeogenesis. It does so by inhibiting the mitochondrial electron chain complex I, which decreases ATP production and increases AMP. This activates AMPK, which, when activated, turns on the catabolic pathways that generate ATP and turns off the anabolic pathway, thereby decreasing the expression of gluconeogenic genes. It also decreases intestinal glucose absorption and increases insulin sensitivity in the skeletal muscle, thus lowering plasma glucose levels [104].

Metformin inhibits tumor growth in cancers that thrive in hyper-insulinemic and hyperglycemic environments. They do so by suppressing hepatic gluconeogenesis and thus reducing glucose and insulin levels. Metformin also leads to the activation of AMPK, which inhibits the mTOR pathway, leading to decreased energy consumption and cell growth [105]. Insulin and Insulin-like Growth Factor-1 (IGF-1) promote cell growth and proliferation. Metformin reduces their levels in the blood by increasing insulin sensitivity and lowering blood glucose levels, suppressing the mTOR pathway, and inhibiting tumor cell proliferation [106].

### 5.2. Metformin and Pancreatic Cancer Prevention

Metformin use in patients with diabetes may prevent the development of pancreatic cancer [107]. A hospital-based case–control study revealed that metformin was associated with a lower risk of developing pancreatic adenocarcinoma than those who took other drugs (odds ratio, 0.38; 95% confidence interval, 0.22–0.69; *p* = 0.001) [108]. Results of a meta-analysis, which included 37 articles, concluded that the summary relative risk (SRR) of overall cancer incidence was 0.73 (95% CI, 0.64–0.83) and mortality was 0.82 (95% CI, 0.76–0.89) [109]. Metformin has also been shown to improve the prognosis of patients with pancreatic cancer. A retrospective study on diabetic patients showed that the 2-year survival rate for the group taking metformin was 30.1% and 15.4% for the group not taking metformin (*p* = 0.004; χ^2^ test) [110]. A meta-analysis was also performed using a wide variety of databases regarding the use of metformin in patients with pancreatic cancer. A total of nine retrospective cohort studies and two randomized controlled trials were included. Significant improvement in survival (Hazard Ratio  =  0.86, 95% CI 0.76–0.97; *p*  <  0.05) was observed in the metformin group compared to the control group. A subgroup analysis concluded that the use of metformin improved survival in resection (HR  =  0.79, 95% CI 0.69–0.91; *p*  <  0.05) and locally advanced tumors (HR  =  0.68, 95% CI 0.55–0.84; *p*  <  0.05), but no improvement was observed in patients with metastasized tumors [34]. A meta-analysis conducted on patients with pancreatic cancer using patient survival as an outcome showed that patients using metformin had a better survival rate than the control group not using metformin (HR = 0.86, 95% CI: 0.76–0.97, *p* = 0.01) [34].

Due to the better side effects of metformin than GLP1- RA and DPP-4 inhibitors, metformin is more favorable for use in pancreatic cancer prevention. A study examined the FDA’s report of adverse effects with the use of the DPP-4 inhibitor, sitagliptin, and the GLP-1 RA, Exenatide [49]. The results showed a 6-fold increase in the odds of pancreatitis in individuals taking sitagliptin (OR, 6.74; 95% CI, 4.61 to 10.00) and a 10-fold increase in those taking Exenatide (OR, 10.68; 95% CI, 7.75 to 15.10). However, a meta-analysis on patients with type 2 diabetes mellitus using DPP-4 inhibitors revealed no association between their use and pancreatitis (Mantel–Haenszel odds ratio with a 95% confidence interval: 0.93 [0.51–1.69]; *p* = 0.82) [111].

### 5.3. Safety and Efficacy of Metformin

Metformin is a well-tolerated, safe, and effective drug with a low risk of cardiovascular events [112]. However, it is contraindicated in patients with severe chronic kidney disease as it can lead to the development of a potentially life-threatening lactic acidosis [113].

## 6. Conclusions

Pancreatic cancer presents at a late stage, has limited therapeutic options, and has high mortality. Its association with diabetes is well established, and pharmacological therapy for diabetes may prevent the development of cancer. Previous research has shown SGLT2 inhibitors to have antiproliferative effects against different types of tumors, including pancreatic carcinoma, due to their blockage of glucose reuptake in cells and suppressing glycolysis via the PI3K/AKT/mTOR pathway. Metformin has been shown to improve the prognosis of pancreatic cancer by inhibiting the mTOR pathway, leading to decreased energy consumption and cell growth and increasing insulin sensitivity, causing decreased blood glucose levels. The association between pancreatic cancer and GLP-1 RA and DPP-4 inhibitors is not as clear, with some studies showing that they may increase the risk of developing pancreatic cancer due to overstimulation of the GLP-1 receptor and an increased risk of pancreatitis. However, subsequent studies on this did not show any increased risk. It is difficult to generalize current studies to the general population, as there may be different effects of drugs depending on the type of cancer and the presence of diabetes. The usage of antidiabetic drugs as adjunctive therapy for pancreatic cancer should be explored. Stratification based on stage and type of tumor would be helpful in applying research to real-life scenarios. As antidiabetic drugs have many benefits in addition to controlling diabetes, they may also be beneficial for cancers beyond the pancreas.

## Figures and Tables

**Table 1 cancers-16-01325-t001:** Articles included in this study and their conclusions regarding association of anti-diabetic drugs with pancreatic cancer.

Drug Class	Mechanism of Action	Suggested Role in Pancreatic Cancer	Literature
GLP-1 Receptor Agonists	Activation of GLP-1 receptors.	Pancreatic Cancer could be a potential long-term side effect of GLP-1 RA use.	Meta-analyses by Aziz et al. [26], Cao et al. [27], Monami et al. [28], and Pinto et al. [29] have reported no increased risk so far.
DPP-4 Inhibitors	Inhibit the enzymatic activity of DPP-4, indirectly prolonging the plasma half-life of GLP-1 and GIP.	Pancreatic cancer could be a potential long-term side effect of DPP-4 inhibitor use.	Meta-analyses by Engel et al. [30], Pinto et al. [29] have reported no increased risk so far. A cohort study by Gokhale et al. [31] reported no elevated risk.
SGLT2 Inhibitors	Lower glucose levels by blocking glucose reabsorption in the proximal convoluted tubules.	SGLT2 inhibitors may decrease pancreatic cancer growth and reduce risk of pancreatic cancer.	A case–control study by Tanaka et al. [32] found decreased risk of pancreatic cancer in diabetic patients taking SGLT2 Inhibitors. Trials in mice by Scafoglio C et al. [33] showed decreased tumor growth and increased tumor necrosis.
Metformin	Suppresses gluconeogenesis in the liver via inhibition of mitochondrial electron chain complex.	Metformin use in patients with diabetes may prevent the development of pancreatic cancer.	A case–control study by Li et al. [34] found metformin was associated with a lower risk of pancreatic cancer A retrospective study by Sadeghi et al. reported better survival rates in patients taking metformin. A meta-analysis by Li et al. [34] found significant improvement in survival in the metformin group.

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
