# Peer review of "Role of SGLT2 Inhibitors, DPP-4 Inhibitors, and Metformin in Pancreatic Cancer Prevention"

_cancers, 2024, doi:10.3390/cancers16071325_

Round 1
Reviewer 1 Report
Comments and Suggestions for Authors
Laeeq et al. present a review about the relationship between use of defined antidiabetic drugs and pancreatic carcinoma risk. Article is interesting and well written. Authors however should be clear that in most cases no causal relationship between OAD use and cancer risk can be found, but rather associations. The only studies with possible causal relationship proof may be randomized placebo controlled clinical trials. On different places of the article authors write what SGLT2 inhibitors and metformin were shown to decrease pancreatic cancer risk. I am not sure what part of literature they cite, result from clinical trial, what part from retrospective studies. I would recommend checking it one more time and do decide when they can write about increased or decreased risk and when about positive or negative associations.
Author Response
Thank you for the feedback. We went through the document again and changed the lines, writing about associations instead of causal relationships.

Reviewer 2 Report
Comments and Suggestions for Authors
Manuscript ID # cancers-2871526
Manuscript Title: Role of SGLTi, DPP4i, and Metformin in Pancreatic Cancer Prevention
Authors: Tooba Laeeq, MD, Maheen Ahmed, Hina Sattar, Muhammad Hamayl Zeeshan and Meher Binte Ali
General Comments
Laeeq T, et al., in their review article entitled, “Role of SGLTi, DPP4i, and Metformin in Pancreatic Cancer Prevention” have discussed the relevant issue of the possible association of certain anti-diabetic drugs/medications (SGLT inhibitors, DPP4 inhibitors and metformin) with the risk of developing and progression of pancreatic cancer. Overall, the manuscript is well written. The theme of the manuscript is appropriate and timely. However, the manuscript must be improved.
Major comments
1) Provide appropriate and clear affiliations for all authors and their university affiliated email id's (which are currently missing) as per Cancers (MDPI) guidelines/template.
2) Provide 'Author contributions' for each author as per Cancers (MDPI) guidelines/template.
3) The corresponding author is not indicated in the manuscript.
4) Figures and tables are key aspects of a review article which are missing in this article. Figures/illustrations for the effect/mechanism of action (either anticancer or tumorigenic/tumor progressive) for each of these classes of anti-hyperglycemic drugs should be provided. Tables depicting the relevant studies, their key findings, molecular changes, and references should be included wherever appropriate.
5) Incretin mimetics or GLP-1/GIP agonists are known to cause pancreatic damage (pancreatitis) and possible malignant transformation of the pancreas, promote the development of medullary thyroid carcinoma (MTC) in mice and possibly in patients with a family history of MTC and 3) thyroid cancers in patients having multiple endocrine neoplasia syndrome type 2 (MEN2). These kinds of drugs are required to provide safety warnings in their information inserts that these drugs are not recommended in patients who have had pancreatitis and should not be used in patients with MTC (or anyone in the family of the patient who has had MTC) or if the patient has multiple MEN2. The authors can thus cover both pancreatic cancer and thyroid malignancies related to this drug in this manuscript. Appropriate figures and tables can be provided.
6) Metformin is one of the most widely studied anti-hyperglycemic effects for its antineoplastic effects. A more elaborate section with the possible benefits of using metformin over the GLP1 RA or DPP4i maybe included.
7) Include sub-sections that include relevant studies with the potential anticancer effects of these drugs and its efficacy in pancreatic/thyroid cancers when used as a monotherapy or in combination with other standard anticancer drugs/natural compounds.
8) Include separate sections with relevant clinical trials/studies/data related to these malignancies for these drugs.
9) Provide future perspectives (raise outstanding questions and what needs to be done in terms of research to address these questions) along with the conclusion.
Minor comments
1) Provide list of abbreviations at the end of the manuscript (before references).
2) Provide funding information, acknowledgement, conflict of interest statement as per Cancers (MDPI) guidelines/template.
3) Use the right formatting for references as per Cancers (MDPI) guidelines/template.
4) Check for typo/spelling errors:
· GLP-1 agonists are sometimes mentioned as GLP 1 RA (line 16) and GLP 1RA (line 17). Uniformity is required.
· DPP4 is randomly mentioned as DPP4 (line 17) and DPP-4 (line 69). Check and make uniform wherever appropriate. Similarly, for GLP1 sometimes indicated as GLP-1. 'glp1' also appears in line 146.
· DPP4 inhibitors are indicated in the title as 'DPP4i' while in the text the abbreviation is given as 'DPP4-I' (line 69).
· TNF-a the 'a' must be replaced by 'a' (greek notation for alpha). Similarly for any other such instances.
· Check for uniformity of the term 'Pancreatic cancer'. The 'P' of Pancreatic and the 'C' of cancer are used randomly in either upper or lower case.
· et al is sometimes indicated as 'et. Al' (lines 141, 148). Check and correct wherever required.
· Check line 289 and 313. Why is this phrase '(mention trials, if any)' mentioned? Delete wherever necessary.
Comments on the Quality of English LanguageTypo errors and spelling mistakes must be corrected. Must be throughly read and corrected.
Author Response
1) Provide appropriate and clear affiliations for all authors and their university affiliated email id's (which are currently missing) as per Cancers (MDPI) guidelines/template.
Some of the authors do not have university affiliated email ids. The full addresses have been provided.
2) Provide 'Author contributions' for each author as per Cancers (MDPI) guidelines/template.
They have been added
3) The corresponding author is not indicated in the manuscript.
Tooba Laeeq has been added as corresponding author
4) Figures and tables are key aspects of a review article which are missing in this article. Figures/illustrations for the effect/mechanism of action (either anticancer or tumorigenic/tumor progressive) for each of these classes of anti-hyperglycemic drugs should be provided. Tables depicting the relevant studies, their key findings, molecular changes, and references should be included wherever appropriate.
A table has been included
5) Incretin mimetics or GLP-1/GIP agonists are known to cause pancreatic damage (pancreatitis) and possible malignant transformation of the pancreas, promote the development of medullary thyroid carcinoma (MTC) in mice and possibly in patients with a family history of MTC and 3) thyroid cancers in patients having multiple endocrine neoplasia syndrome type 2 (MEN2). These kinds of drugs are required to provide safety warnings in their information inserts that these drugs are not recommended in patients who have had pancreatitis and should not be used in patients with MTC (or anyone in the family of the patient who has had MTC) or if the patient has multiple MEN2. The authors can thus cover both pancreatic cancer and thyroid malignancies related to this drug in this manuscript. Appropriate figures and tables can be provided.
A paragraph on thyroid cancer has been added.
6) Metformin is one of the most widely studied anti-hyperglycemic effects for its antineoplastic effects. A more elaborate section with the possible benefits of using metformin over the GLP1 RA or DPP4i maybe included.
Another paragraph on metformin has been added
7) Include sub-sections that include relevant studies with the potential anticancer effects of these drugs and its efficacy in pancreatic/thyroid cancers when used as a monotherapy or in combination with other standard anticancer drugs/natural compounds.
Subsections include introductions to drugs, their tumorigenesis, role in pancreatic cancer, and safety profile. We have included these to make the article interesting to read and easy to follow.
8) Include separate sections with relevant clinical trials/studies/data related to these malignancies for these drugs.
We have provided citations and have included trials or meta-analyses of trials that were done previously.
9) Provide future perspectives (raise outstanding questions and what needs to be done in terms of research to address these questions) along with the conclusion.
It has been added to the conclusion.
Minor comments
1) Provide list of abbreviations at the end of the manuscript (before references).
The list has been added.
2) Provide funding information, acknowledgement, conflict of interest statement as per Cancers (MDPI) guidelines/template.
These parts have been added.
3) Use the right formatting for references as per Cancers (MDPI) guidelines/template.
The references have been edited
4) Check for typo/spelling errors:
- GLP-1 agonists are sometimes mentioned as GLP 1 RA (line 16) and GLP 1RA (line 17). Uniformity is required.
Changes have been made.
- DPP4 is randomly mentioned as DPP4 (line 17) and DPP-4 (line 69). Check and make uniform wherever appropriate. Similarly, for GLP1 sometimes indicated as GLP-1. 'glp1' also appears in line 146.
Changes have been made.
- DPP4 inhibitors are indicated in the title as 'DPP4i' while in the text the abbreviation is given as 'DPP4-I' (line 69).
Changes have been made
- TNF-athe 'a' must be replaced by 'a' (greek notation for alpha). Similarly for any other such instances.
Changes have been made
- Check for uniformity of the term 'Pancreatic cancer'. The 'P' of Pancreatic and the 'C' of cancer are used randomly in either upper or lower case.
Changes have been made
- et al is sometimes indicated as 'et. Al' (lines 141, 148). Check and correct wherever required.
Changes have been made
- Check line 289 and 313. Why is this phrase '(mention trials, if any)'mentioned? Delete wherever necessary.
The lines have been deleted
Reviewer 3 Report
Comments and Suggestions for Authors
This study reviewed the clinical association between pancreatic cancer and antidiabetic drugs (SGLT2 inhibitors, GLP 1RA, DPP4 inhibitors, and biguanides). The authors showed that GLP 1RA and DPP4 inhibitors might increase the risk of development of pancreatic cancer, but DPP4 and Metformin could decrease it and improve the survival. This paper is well-written using recently published literature, and easy to follow.
Author Response
Thank you for your feedback